# The creation of electric wind due to the electrohydrodynamic force

Sanghoo Park [1], Uros Cvelbar[2], Wonho Choe [1,3] & Se Youn Moon[4]

Understanding the interactions between ionized matter and neutral particles is a prerequisite for discovering their impact on natural phenomena. One such phenomenon is the electric wind, which supposedly occurs due to the charged particle–neutral coupling in systems of weakly ionized gases, but this mechanism remains unclear. Here, we report direct evidence that electric wind is caused by an electrohydrodynamic force generated by the charged particle drag as a result of the momentum transfer from electrons/ions to neutrals. The model experiment is based on a pulsed plasma jet as a source of weakly ionized gases generated in the helium gas at atmospheric pressure using Schlieren photography. Studying the helium gas flow trajectories at different discharge parameters allows one to distinguish between the effects of streamer propagation or space charge drift causing the electric wind as well as to determine the role of electrons and (positive) ions in wind generation.

[1] Department of Physics, Korea Advanced Institute of Science and Technology (KAIST), 291 Daehak-ro, Yuseong-gu, Daejeon 34141, Republic of Korea. [2] Jožef Stefan Institute, Jamova cesta 39, SI-1000 Ljubljana, Slovenia. [3] Department of Nuclear and Quantum Engineering, Korea Advanced Institute of Science and Technology (KAIST), 291 Daehak-ro, Yuseong-gu, Daejeon 34141, Republic of Korea. [4] Department of Quantum System Engineering, Chonbuk National University, 567 Baekje-daero, Deokjin-gu, Jeonju 54896, Republic of Korea. Correspondence and requests for materials should be addressed to W.C. (email: wchoe@kaist.ac.kr)

Collisional coupling (that is momentum and energy exchange) between charged particles and neutral particles (c–n) can significantly impact any natural phenomena involving weakly ionized gases. Solving the hydrodynamic problem of such phenomena has been of paramount importance for centuries and requires expertise in a wide range of disciplines. After the first observation of c–n coupling by F. Hauksbee in the 1700s, this phenomenon rapidly became a popular scientific subject and attracted the attention of notable scientists, including of M. Faraday and J. C. Maxwell[1]. In recent decades, state-of-the-art experimental approaches with knowledge-intensive instruments have enabled the discovery of numerous unexpected pieces and evidence that relate this natural phenomenon and its relevant mechanisms to c–n coupling. In planetary or stellar atmospheres (in which electrons, ions, and neutrals coexist in sufficient densities), including those of the Earth, the ion–neutral coupling profoundly affects ion and neutral particle kinetics and their properties[2–5]. The neutral drag induces an electric field through the charge separation between ions and electrons under the geomagnetic field, while the ion drag resulting from the electric and geomagnetic fields exerts a force on neutrals[2–4]. This charged particle–neutral coupling is also applied in practical engineering. One of the well-known cases of this effect is the electric wind (also called ionic wind), which is created by an electro-hydrodynamic (EHD) force in electrically charged fluids such as weakly ionized plasma and ionic solutions. The electric wind generally arises in non-thermal air discharges, such as DC corona discharge and surface dielectric barrier discharge[6–8]; in principle, the charged species in weakly ionized air are accelerated by the electric field and transfer their momentum to neutrals via high-frequency collisions. Because of the possibility of producing a gas flow using only electrical energy without mechanical generation, these types of discharges have been investigated and applied in aerodynamic applications as alternative flow controllers.

In addition to the aforementioned electric wind actuators, the extraordinary gas flow is manifested also in plasma jets operating at atmospheric pressure[9–11]. In the jets, where a horizontally ejected neutral helium gas flow is used, the gas flow trajectories, or so-called free jet boundary, are typically bent upward due to the buoyant force. The flow trajectories are lowered toward the horizontal plane when the neutral gas flow is faster. Nevertheless, when the electrical discharge, or so-called plasma, is generated inside the same flow, the gas speed of the flow increases. This is observed from the flow trajectories, which is additionally further lowered toward the horizontal plane of the flow. Therefore, this phenomenon resembles the increase in the flow rate or gas speed. For example, the flow rate in a helium plasma jet at 0.2 standard liters per minute (slpm) corresponds to a neutral helium gas flow rate of approximately 0.47 slpm without discharge[10].

Plasma jet devices are typically designed with dielectric tubing and a discharge electrode, either in the form of a single electrode inside the tubing or two ring electrodes outside the tubing. The electrodes are used to ionize a flowing noble gas, which is then released into the ambient air or toward a target at atmospheric pressure conditions. Such devices differ from a conventional plasma apparatus, where the plasma is typically confined by electrodes or generated inside the chamber at a reduced pressure. However, the plasma jet characteristics are extremely sensitive to changes in the gas structure, the type, and its flow dynamics (for example, laminar-turbulent transition, gas flow speed, and

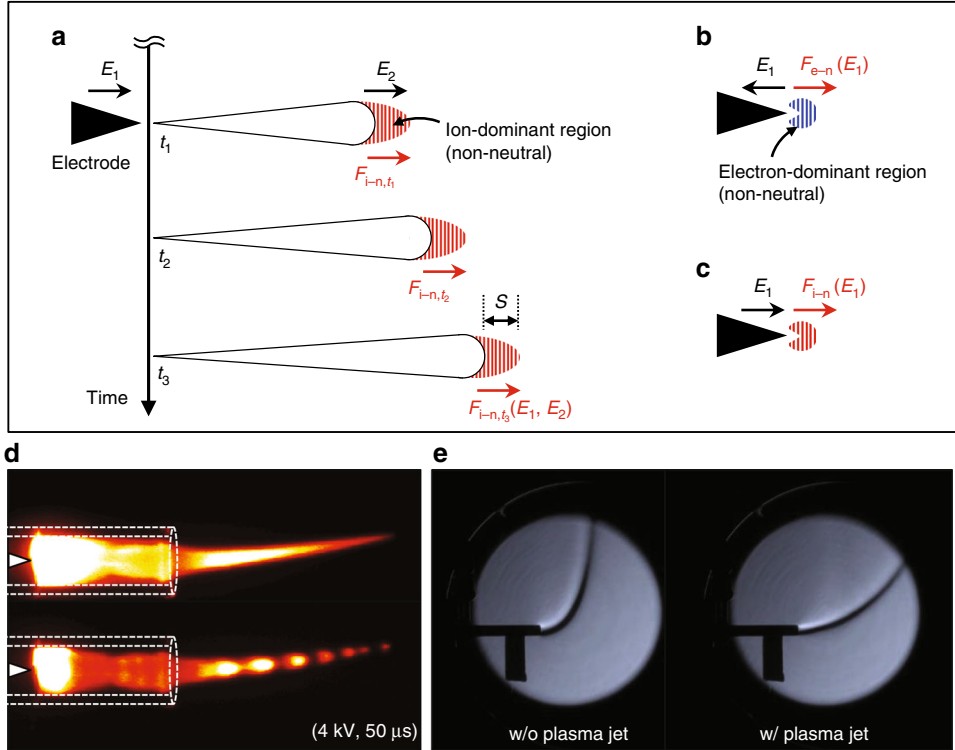

**Fig. 1** A simple illustration of EHD force generation via different mechanisms. The schematic of EHD force generation by **a** positive pulsed streamer propagation driven by (positive) ion and the presented space charges behind the moving front, where $E_1$ and $E_2$ represent external and internal electrical fields, respectively. The cases of EHD force generation by **b** electron drift in negatively pulsed plasma jets or **c** ion drift in positively pulsed plasma jets are presented. **d** A plasma jet image in continuous mode and composite image of nanosecond-resolved images are presented along with a schematic of the experiment with the glass tubing and electrode. **e** The jet of free gas flow without plasma has a flow trajectory or free jet boundary, which is imaged by Schlieren photography, and this trajectory changes when the plasma discharge is turned on, thereby providing information on the flow parameters. The glass tubes with an inner diameter of 3.5 mm are used as substitute for scale bars for all images

trajectory). Moreover, the jet characteristics are also dependent on the discharge parameters (such as discharge voltage, current, pulse width, and tubing diameter) and discharge type[12,13]. A good example of the electric wind affecting a plasma jet is the multi-array plasma jet system used for large-area surface treatment. Mutual interactions between neighboring pulsed plasma streamers induce the change in the neutral gas flow repulsively[11,14]. In addition, the electric wind can cause other effects, such as modifications of the streamer generation and propagation, thus altering the neutral gas density and/or the efficient transport of long-lived chemical species with lifetimes ranging from milliseconds to minutes (such as metastable helium atoms, ozone, and nitrogen oxides) due to the enhancement of the gas flow. Many efforts have been made toward discovering the c–n coupling associated with the electric wind for cases such as atmospheric-pressure plasma jets[9,10,15]. However, the current experimental evidence and analyses of these electric winds are not sufficient to unravel its true origin.

Here, we show the origin of the electric wind through a simple model experiment, which is designed using a helium flow and a μs-pulsed plasma jet. Changes in the helium jet trajectory depending on pulse height and width allow to simply demonstrate the major contribution of EHD force on the electric wind in a plasma jet. This phenomena is directly connected with the densities of charged species moving and interacting with neutrals that create a plasma head and electrons or ions coupling with neutrals behind generating the space charge. As such, we provide a direct observation of charged particle–neutrals couplings.

## Results

**Theoretical background and model.** The EHD force in gaseous discharges or the speed of the electric wind, which changes the gas dynamics in a plasma jet, are easily estimated with current models[6,7]. The mechanism underlying the electric wind in a plasma jet is schematically illustrated in Fig. 1a–c along with its two major mechanisms: the space charge drift and the streamer propagation[7,16]. The EHD force per unit volume $F_{i,e-n}$ in a weakly ionized gas can be approximated as

$$F_{i,e-n} \approx q(n_i - n_e)E = \frac{j_i}{\mu_i} - \frac{j_e}{\mu_e}, \quad (1)$$

where $q$ is the elementary charge; $n_i$ ($n_e$), $j_i$ ($j_e$), and $\mu_i$ ($\mu_e$) are the number density, the current density, and the mobility of ions (electrons), respectively; and $E$ is the electric field. Equation (1) indicates that the electric wind is generated in the non-neutral region where $n_i \neq n_e$. In the case of a cathode-directed (positive) streamer (that is a plasma bullet or pulsed plasma streamer) propagation, the electric wind is induced as illustrated in Fig. 1a. In the non-neutral region in which the net EHD force is non-zero, the force is placed on the leading front or head of the pulsed plasma streamer. Moreover, the contribution of electrons can be neglected in Eq. (1) because ions are abundant in the moving streamer head, that is, $n_i \gg n_e$. The force is governed by the externally applied electric field ($E_1$) and the localized (internal) electric field ($E_2$) built up by the streamer head. To obtain the time-averaged force, the effective time $\delta t$, which is defined by the length of the streamer head (denoted by $S$ in Fig. 1a) divided by the streamer propagation speed, during which the EHD force is exerted on the ions in the moving streamer head has to be considered. The time-averaged expression of Eq. (1) for the moving streamer force can then be written as

$$\overline{F}_{str} = \frac{1}{T} \int_0^{\delta t} \left(\frac{j_i}{\mu_i}\right)_{str} dt \approx \frac{1}{T} \left(\frac{j_i}{\mu_i}\right)_{str} \delta t, \quad (2)$$

where $T$ is the pulse repetition time. Because of the mass difference between the electrons and ions and the charging of dielectric surfaces around the plasma, the non-neutral region can also exist for some time after the moving streamer disappears. During this period, the EHD force is sustained by the residual space charges. The time-averaged EHD force from the space charge can then be given as

$$\overline{F}_{SC} = \frac{1}{T} \int_0^{\tau} \left(\frac{j_i}{\mu_i} - \frac{j_e}{\mu_e}\right)_{SC} dt, \quad (3)$$

where $\tau$ is the pulse width. $\overline{F}_{SC}$ includes electron- and ion-related terms because of both contributions, from electrons during the negative-polarity period (see Fig. 1b) and from ions during the positive-polarity period (see Fig. 1c), which should be considered in the case of bipolar voltages applied to the electrode. Here, it is worth noticing that the negative ions (such as $O_2^-$ and $O_3^-$) can also affect the electric wind in oxygen-added plasmas; however, in our case study, this can be neglected since negative ions are rare in pure helium plasma jets[13,17].

For simplicity, the problem is treated as one dimensional along the streamer propagation direction since the propagation distance is much larger than the streamer width, as seen in Fig. 1d. This holds in our experiments for a majority of cases. The analytic expression for the electric wind speed is then obtained from the relation between the dynamic pressure $p$ and the total EHD force:

$$p = \int_0^L (\overline{F}_{str} + \overline{F}_{SC}) dx$$
$$\approx \int_0^L \left[ \frac{1}{T} \left(\frac{j_i}{\mu_i}\right)_{str} \delta t + \frac{1}{T} \int_0^{\tau} \left(\frac{j_i}{\mu_i} - \frac{j_e}{\mu_e}\right)_{SC} dt \right] dx, \quad (4)$$

where $L$ is the length of the plasma jet. This expression is directly related to the electric wind speed $v_E$ as $p = \rho_G v_E^2/2$, where $\rho_G$ is the background neutral mass density. Thus, the electric wind speed is written as

$$v_E = \sqrt{\frac{2}{\rho_G} \int_0^L \left[ \frac{1}{T} \left(\frac{j_i}{\mu_i}\right)_{str} \delta t + \frac{1}{T} \int_0^{\tau} \left(\frac{j_i}{\mu_i} - \frac{j_e}{\mu_e}\right)_{SC} dt \right] dx}. \quad (5)$$

Although the electric wind speed is expressed in a rather simple manner, as in Eq. (5), accurately measuring $j_{e,i}$ (or $n_{e,i}$) and $\delta t$ is not straightforward due to the absence of appropriate diagnostic methods. Reports on correlations between the electric wind speed and the plasma conditions in terms of the voltage pulse width and height, which are linked to plasma properties, have yet to be made available. Moreover, there is no convincing evidence regarding the major mechanism for the generation of electric wind in atmospheric pressure plasma jets, which is created either by the streamer propagation or the space charge drift.

To resolve this problem, a simple model experiment was designed using a helium flow and a μs-pulsed plasma jet generated at different voltage pulse widths and heights, and Schlieren photography was used to image the neutral gas flow ejected from a plasma jet. The helium gas flow channel followed a parabolic trajectory when spreading into the ambient air because of the buoyant force. Thus, based on changes in the gas flow trajectories caused by the electric wind, we can clearly see that the operation parameters affect the electric wind in the plasma jet (see Fig. 1e). As such, they can be used to elucidate the predominant mechanism underlying the electric wind. Because of the low Reynolds numbers, the gas flow was laminar under all of our plasma conditions. Here, it is worth noting that, in this experiment, the inaccuracies in the measurements induced by changes in the buoyant force due to the gas heating are neglected because the gas temperature remained almost constant across all

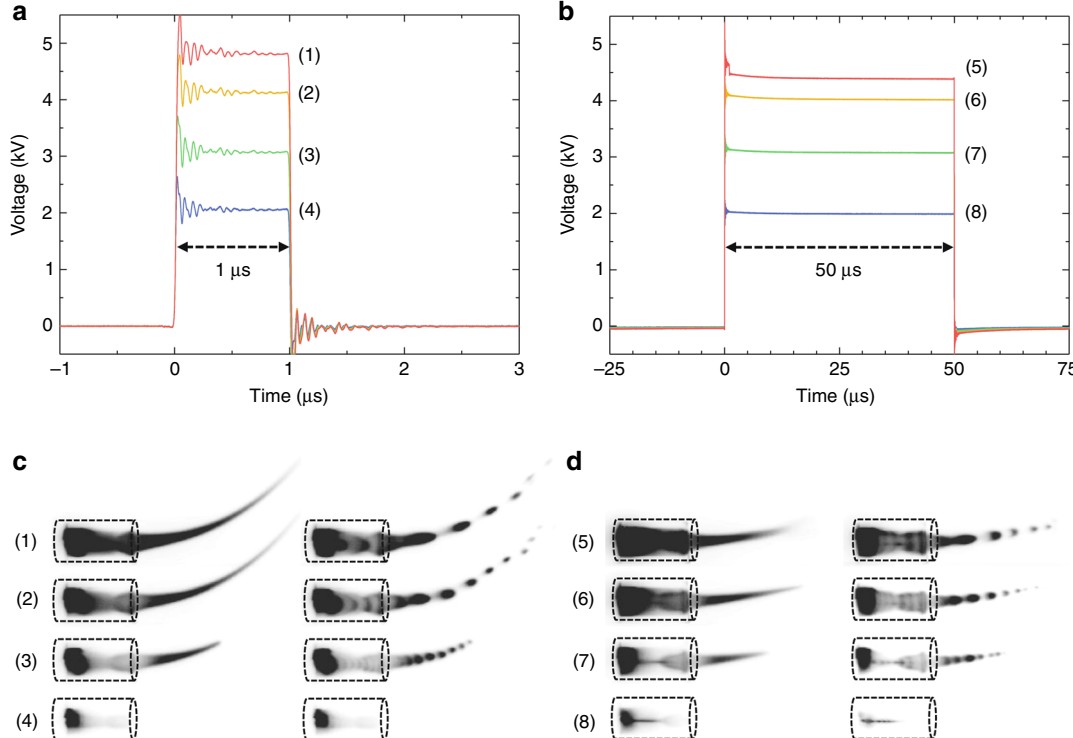

**Fig. 2** Voltage waveforms with different pulse widths and heights. Voltage pulses having pulse widths of **a** 1 μs and **b** 50 μs at different pulse heights applied to the electrode. Time-integrated (left column) and composite nanosecond-resolved (right column) plasma emission images at **c** 1 μs and **d** 50 μs; specific voltage heights are denoted by numbers in this figure. The glass tubes with an inner diameter of 3.5 mm are used as substitute for scale bars for all images

operational conditions. Moreover, the neutral particle density also remained constant in the plasma jet, which is representing a weakly ionized gaseous discharge ($n_e/n_a < 10^{-7}$, where $n_a$ is the number density of neutral atoms).

**Negligible effect of gas heating on the electric wind**. Gas heating would lead to inaccurate analyses of electric wind generation and is frequently the most significant problem for observing the EHD force as well as in determining its mechanisms. Therefore, many researchers have underestimated this phenomena in their studies[18–20]. Most frequently, this occurs because the gas heating is regarded as a major cause of increasing the gas speed and changing the flow structure, whereas the EHD force has no influence since the time-averaged EHD force is negligible[18,19]. However, although the ejected gas speed increased by approximately 25% in the heated gas case, the flow pattern remains the same[20]; considering the continuity of the mass flow rate, the increased buoyancy force due to the 25% lower density of the heated helium should compensate for the 25% increase in the outlet speed. Consequently, the increase in the neutral gas temperature via electron–neutral collisions in the plasma jet does not drastically affect the gas dynamics, including the flow trajectory. However, minor changes in the flow structure may still occur.

Because the gas temperature is constant in our experiment, as described below, the flow structure results obtained by Schlieren photography can be treated without any compensation for gas heating. To estimate the gas temperature and confirm its changes under different driving conditions non-invasively, the emission spectra of the nitrogen molecular ion $N_2^+\left(B^2\Sigma_u^+ - X^2\Sigma_g^+\right)$ in the plasma jets were collected near the tube exit using a spectrometer (Acton Research SpectraPro 750). Surprisingly, no difference in the spectral distribution of the $N_2^+$ molecular band was observed among all the studied cases (see Supplementary Note 1 and

Supplementary Fig. 1a). This indicates that the rotational temperature of $N_2^+$ is independent of the pulse width and pulse height. Therefore, the neutral gas temperature and thus the density were not modified under the operating conditions. The experimentally measured $N_2^+$ spectrum and the well-fitted synthetic spectrum (see Supplementary Fig. 1b) indicated a rotational temperature of 340 K. From these results, we concluded that the predominant mechanism responsible for changes in the helium flow trajectory is not the gas heating but rather the electric wind caused by the EHD force.

**Operation parameters and plasma plume structures**. To investigate the contribution of each force introduced above ($\overline{F}_{str}$ and $\overline{F}_{SC}$), two different sets of plasma were produced with different pulse heights (2–5 kV) and pulse widths (1 and 50 μs), as presented in Fig. 2a, b. The pulse repetition rate was fixed at 10 kHz, and thus, the corresponding duty cycles were 1% and 50%, respectively. In both cases, the cathode-directed (positive) streamers were produced and propagated during the rise phase of the voltage pulse. The lifetime (that is, the period between its initiation and collapse) of the pulsed streamer was approximately 1 μs, and therefore, the plasma streamer existed throughout the on-cycle in the case of 1 μs pulse width. Although the secondary anode-directed streamers are formed in the voltage fall phase, they play an insignificant role in generating electric wind in positive-polarity pulsed plasma jets and therefore are neglected. However, as discussed later, these anode-directed (negative) streamers also affect the EHD force via momentum transfer from electrons to neutrals in the negative-polarity pulsed plasma jets. In the cathode-directed (positive) streamers, the momentum transfer occurs from ions to neutrals in the moving streamer head and generates a jet with different lengths under different pulse widths. This is presented in the left and right columns of Fig. 2c, d

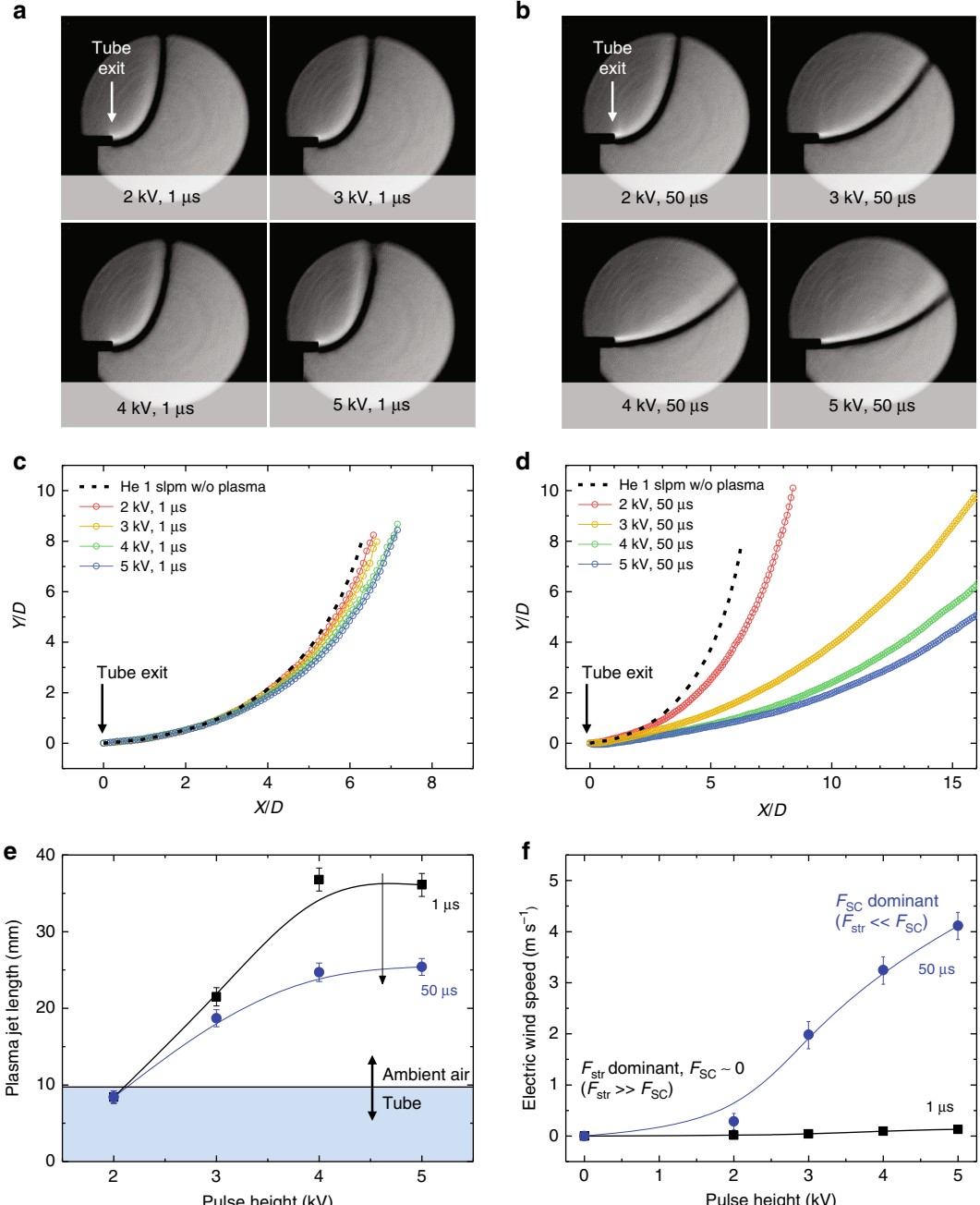

**Fig. 3** Variation in the gas flow trajectory. Schlieren images of helium flow ejected from the plasma jets at different applied voltages with constant pulse widths of **a** 1 μs and **b** 50 μs and **c**, **d** corresponding centerlines of helium flow, without plasma (black dashed lines) and in the presence of helium plasma jets generated at different pulse heights. The corresponding **e** plasma jet total lengths measured over the curvature from the electrode (as detailed in Supplementary Fig. 4) are presented with **f** the calculated electric wind speeds. The glass tubes with an inner diameter of 3.5 mm are used as substitute for scale bars for all images. Data in **e**, **f** are shown as mean ± s.d. for all panels; $n = 5$

with the time-integrated emission of the plasma jet and a composite sequence of ns-resolved images, respectively. These composite images are constructed by overlaying multiple frames obtained using an intensified charge-coupled device camera (Andor DH312T) with an exposure time of 20 ns and a time interval of 60 ns. A difference in the bending shape of the plasma plumes and their lengths between the two pulse widths is evident from these figures. In principle, the generated pulsed plasma streamer is guided along the helium gas flow, and thus, the time-integrated luminous plasma plume appears to bend upward. The exact trajectory of the helium flow ejected from the end of the

dielectric tubing is observed from Schlieren images. The roles of the operating parameters on electric wind generation can be intuitively interpreted based on the analytic solutions given by Eqs. (2, 3). As the amplitude of the voltage pulse increases, the externally applied electric field as well as the electron and ion densities also increases, thereby resulting in the increasing localized electric field, propagation speed of the plasma streamers, and total plume length. The last two cases are observed in Fig. 2c, d. Consequently, the EHD force activated in this case by the positive streamer is increased as the voltage amplitude increases at the constant pulse width. Similarly, an increase in the time-

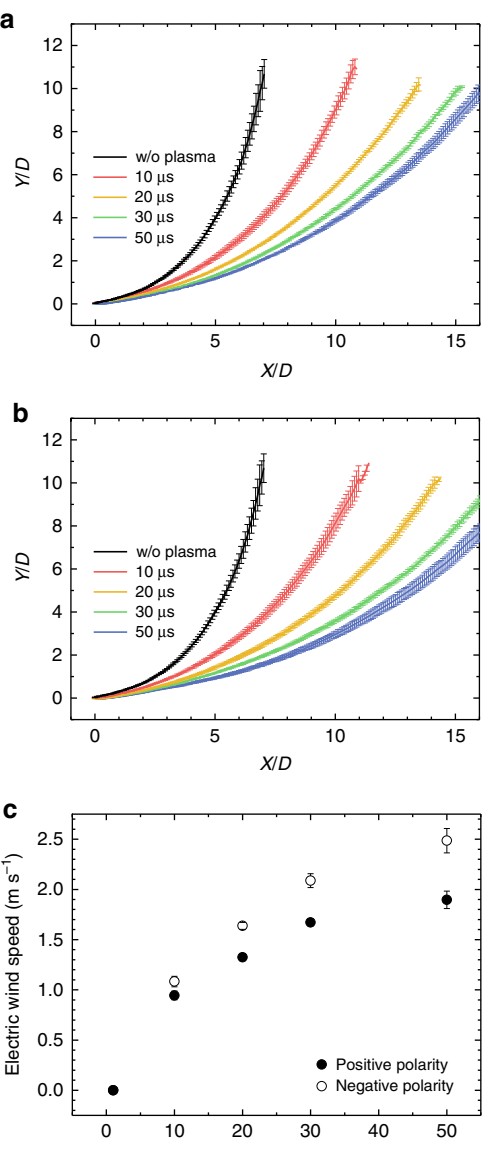

**Fig. 4** Positive- and negative-polarity voltage pulses and their effects on the flow trajectory. The helium flow trajectories for **a** positive-polarity and **b** negative-polarity pulsed plasma jets with pulse widths of 10–50 μs generated at 3 kV are presented. Corresponding results are generated with the voltage and current waveforms from Supplementary Fig. 5. The differences in the electric wind speeds for both polarity cases are depicted from **c**, which are the result of space charge species remaining behind the moving streamer head. Data are shown as mean ± s.d. for all panels; $n = 5$

averaged EHD force is expected as the pulse width increases when the net space charge remains constant, as in Eq. (3).

**The origin of electric wind**. To finally unravel the identity of the electric wind, the amplitude effect of the pulsed voltage on the gas flow trajectory at different pulse widths was examined, as presented for cases of interest in Fig. 3. The centerline of the flow trajectory is plotted in $(X, Y)$ coordinates with the dimensionless variables $X/D$ and $Y/D$, where $D$ is the diameter of the dielectric tubing. The tube nozzle exit is located at the origin (0, 0). The Schlieren images of helium plasma jets and the corresponding centerlines of helium flow with different pulse heights of 1 μs and 50 μs are displayed in Fig. 3a–d, respectively. These photographs

and graphs reveal that the gas flow trajectories are bent upward by the vertical buoyant force. Compared to the trajectory of the pure helium gas flow without plasma (denoted by black dashed curves), the gas flow becomes relatively horizontal in the presence of plasma and exhibits a smaller angle relative to the horizontal plane. This behavior is attributed to the increased gas speed caused by the electric wind. As presented in Fig. 3e, the plume length increases as the voltage increases in both cases (that is, when the pulse widths are 1 μs and 50 μs). However, the helium trajectories of the two cases differ. When the pulse width is 1 μs, the helium flow trajectories with and without plasma are similar, as depicted in Fig. 3c. Furthermore, there are no noticeable differences in the gas speed along the jet axis at different electrode voltages (Fig. 3f), although the plasma streamer propagates farther as the voltage increases (Figs. 2c and 3e). As mentioned before, the pulsed plasma streamer lifetime is approximately 1 μs, and thus, the contribution of the space charges to the EHD force is negligible (that is, $\overline{F}_{SC} \sim 0$). The slight difference in the flow trajectory and thus the flow speed as the electrode voltage was varied suggests that $\overline{F}_{str}$ is also small.Please check the level 2 heading "The origin on the electric wind" under Results section. Should it be origin on or origin of in the same heading.The authors appreciate this notice. The sentence "The origin on the electric wind" is now corrected as follows: "The origin of electric wind"

Unlike the 1 μs case, the bending angle of the gas flow decreases dramatically as the voltage increases when the pulse width is 50 μs, as presented in Fig. 3d. In this case, $\overline{F}_{str}$ remains small, as is the case for the 1 μs pulse width. However, the contribution from the space charges becomes important and enhances the gas flow speed. Consequently, the electric wind is generated by the drift of space charges, whereas the contribution of plasma streamer propagation to the total time-averaged EHD force is negligible (that is, $\overline{F}_{str} \ll \overline{F}_{SC}$). The increase in the electric wind speed is enormous and an order of magnitude higher, as seen in Fig. 3f. The electric wind increases from 15-times (0.02–0.29 m s$^{-1}$) to 50-times (0.04–1.98 m s$^{-1}$) at pulse heights of 2 kV and 3 kV, respectively. Despite this, our results do not indicate that the plasma streamer is unimportant because the characteristics of plasma streamers are related to the concentration of residual space charges, which affect $\overline{F}_{SC}$, especially after the plasma streamer collapses. Here, the terms 'space charges' and 'streamer propagation' are deliberately mentioned as generalized terms underlying basic mechanisms, where more complex systems for the generation of jets and reactions occurring inside helium plasma and ambient air are described in Supplementary Note 2 and Supplementary Figs. 2 and 3.

**The role of charged particles in the electric wind**. To figure out how the electric wind is affected by the charged species generated in atmospheric pressure plasmas during the propagation of a streamer, experiments were performed for positive- and negative-pulsed voltage periods. To confirm the roles of the pulse width and polarity in electric wind generation, changes in the flow trajectories were investigated by controlling the pulse width from 10 to 50 μs with a fixed pulsed height of 3 kV. The results are presented in Fig. 4, where the details on the discharge parameters are presented in Supplementary Fig. 5. The discharges occurred during the rise and fall phases of the voltage pulse in both polarity cases. Because the pulsed plasma streamer propagation time (~1 μs) is much shorter than the pulse width (10–50 μs), the plasma conditions, including the plume length, discharge current, streamer propagation speed, and neutral gas temperature, remained almost constant in the pulse width range from 10 to 50 μs. Therefore, by directly comparing the flow trajectories with

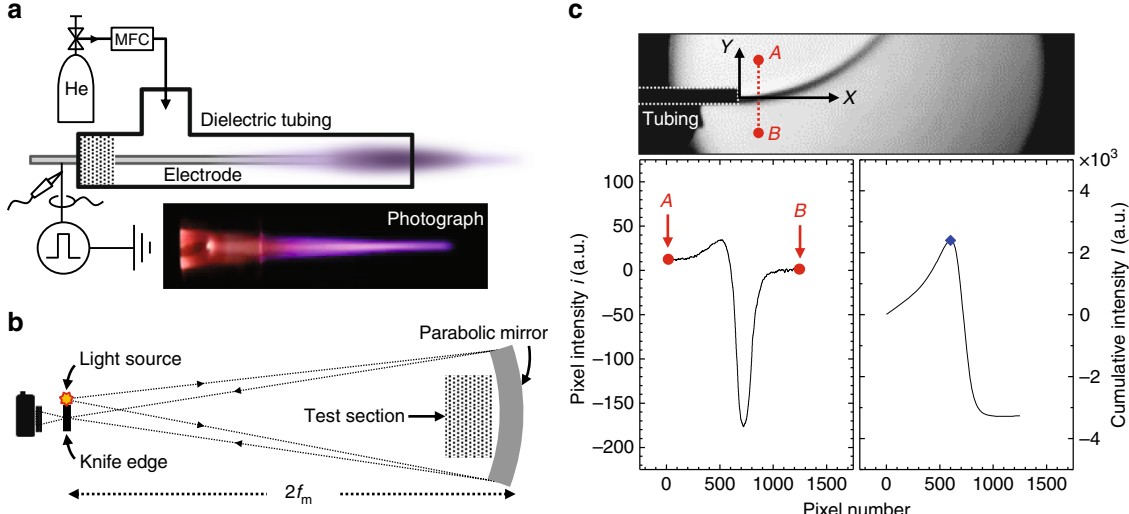

**Fig. 5** Experimental apparatus for atmospheric-pressure plasma jet and Schlieren photography. **a** Schematic of a plasma jet actuator and relevant setup. The μs-pulsed plasma jet was operated by applying a high-voltage pulse to the pin electrode in the presence of helium gas. A real image of a helium plasma jet is also shown in the inset of the figure. **b** Schematic of the custom-made Schlieren imaging system (single-mirror type) and **c** an example method for extracting the centerline of the helium flow from Schlieren images. The left and right graphs show the pixel intensity and cumulative intensity of a given Schlieren image, respectively

different pulse widths, the force acting on the neutrals after the plasma streamer disappears can be estimated. As expressed in Eq. (5), the gas flows are expected to become more horizontal as the pulse width (or duty cycle) increases. Helium gas flow trajectories are plotted in Fig. 4a, b at pulse widths of 10, 20, 30, and 50 μs for both polarity cases. As depicted in the figures, electric wind occurs with both the positive pulse and the negative pulse. In addition, the effect becomes increasingly significant as the pulse width becomes wider at the same voltage in both cases. In the case of a negative pulse, the external electric field becomes zero after the cathode-directed streamer is generated. Therefore, the EHD force cannot be exerted by drifting positive ions in moving head and in tail but can be by the remaining electrons in tail, or space charge, under the negative pulse period (that is, after the anode-directed streamer is propagated). These electrons then induce changes in the gas flow. A similar result holds for the positive pulses with electrons in moving head and remaining electrons in the tail, as discussed above. However, judging by the results for the electric wind speed in Fig. 4c, the depicted speeds for the negative polarity are higher than for the positive polarity. As indicated previously, the electric wind observed in the plasma jet is mainly induced by the drift of space charges (that is, $\overline{F}_{str} \ll \overline{F}_{SC}$). Thus, in the case of negative polarity, there are more space charges behind the moving streamer head than in positive-polarity-generated streamers. This is in accordance with previous observations and the theory of cathode–anode discharges, where researchers have found a hollow-shape radial distribution of excited plasma species in time-averaged cathode-directed streamers, which correspond to the lower optical emission rates of the jets[12,13]. In contrast to the anode-directed streamers, the time-averaged distributions of excited plasma species are more continuous with increasing inner radial distribution. Accordingly, the absolute densities of space charges that contribute to the EHD force are higher for the negative polarity discharges at comparable measurement and discharge parameters, thereby resulting in the higher speed of electric wind. All this is verified and observed as presented in Supplementary Fig. 6, which is schematically analyzed in greater detail in Supplementary Note 2. However, the EHD force does not continually increase as the pulse width increases because the concentration of space charges decays

exponentially over time after a luminous plasma channel becomes dark.

To conclude, the mechanism of electric wind creation is revealed through a study case with a neutral helium gas flow and generated pulsed plasma jet at atmospheric pressure. Our findings indicate that the contribution of the moving plasma streamer to EHD force generation is negligible (as confirmed by similar gas flow trajectories with and without 1-μs pulsed plasma). Second, the EHD force is mainly caused by the residual space charges after the plasma streamer propagates and collapses. The electric wind becomes more significant as the pulse width increases (to some extent) at a given pulse voltage and as the pulse height (that is, electrode voltage) increases when the pulse width exceeds the bullet lifetime. This is directly connected with the densities of charged species moving and interacting with neutrals that create a plasma head and electrons or ions coupling with neutrals behind generating the space charge. This is the first clear report of a separate observation of charged particle–neutrals couplings. Moreover, this report provides fundamental knowledge about the electric wind generated by the EHD force and can serve as a basic reference for understanding the coupling between charged particles and neutrals in any plasma, ranging from plasma processing, fusion, and astrophysics to space propulsion.

## Methods

**Pulsed plasma jet system**. The schematic of an experimental system used to produce helium plasma jets is shown in Fig. 5a. A single-electrode plasma jet, which is one of the simplest and most widely used configurations, was employed for this work. A copper wire covered by a T-shaped fused silica tubing with an inner diameter of 3.5 mm was used as a powered electrode. Helium gas (99.999% purity) was supplied by a calibrated mass flow controller (MKS Type 1179) at a constant flow rate of 1 slpm through the dielectric tubing. A high-voltage DC pulse system consisting of a pulse generator (Directed Energy PVX-4110), a high-voltage amplifier (Trek 10/10B-HS), and a signal generator (Agilent 33512B) was connected to the electrode. The voltage and current waveforms were obtained by a high-voltage probe (Tektronix P6015A) and a Rogowski coil current probe (Pearson 4100).

**Schlieren imaging system and imaging method**. The helium neutral gas flow trajectories were monitored by a Schlieren imaging system, which consisted of a digital camera (Canon EOS 500D), a knife edge, a light source, and a parabolic mirror (Edmund Optics 32-071-533), as illustrated in Fig. 5b. The plasma jet

source described above was located at the test section. The parabolic mirror had an effective focal length $f_m$ of 36 inches and a diameter of 6 inches. A white light-emitting diode was used as the point light source. Because the contribution to the refractive index from the neutral atoms is dominant in weakly ionized plasmas in the visible spectral range, a Schlieren image with a visible light source was used to monitor the density gradient of the neutrals. The diffraction caused by the electrons was considered negligible. The direction of the knife edge (that is, either vertical or horizontal) depends on the direction of the neutral particle density gradient. In this work, the density gradients were oriented in both directions (that is, perpendicular and parallel to the dielectric tubing) because the buoyant force acting on the helium gas ejected horizontally from the tubing caused the flow channel to bend upward. As a result, two knife edges were installed to block both directions. All Schlieren images used in this paper were taken with an exposure time of 1/40 s, f/40, and ISO 100. All measurements were revivified and reproduced at least 10 times.

**Determination of the flow centerline from the Schlieren image**. The Schlieren images taken by the imaging system introduced above were analyzed as follows. Because the intensity profile of a Schlieren image is the result of neutral density gradients perpendicular to the jet-axis[21], the location of the maximum value of the cumulative intensity $I$ distribution of pixel intensity $i$ perpendicular to the jet-axis represents the centerline of the flow channel. For example, the spatial profiles of $i$ and $I$ at constant $X_1$ are extracted from each Schlieren image, and the center of the flow channel (depicted by a blue dot in the plot) is determined, as presented in Fig. 5c. The cumulative intensity distribution along the $Y$ direction is given as $I(X_1, A \le Y_1 \le B) = \sum_{Y=A}^{Y_1} i(X_1, Y)$, where $X_1$ is the specific distance (or pixel number) from the end of the tubing, defined as the origin of the $(X, Y)$ coordinate system. The values of $A$ and $B$ are the certain positions in the flow channel.

Other details on methods and result, including any associated references, are available in Supplementary Methods.

**Data availability**. The authors declare that the data supporting the findings of this study are available within the paper and its Supplementary Information file. All additional raw and derived data that support the plots within this paper and other findings of this study are available from the corresponding author upon reasonable request.

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

## Acknowledgements

The authors acknowledge financial support from the R&D Programs of 'Plasma Advanced Technology for Agriculture and Food (Plasma Farming)' of the National Fusion Research Institute of Korea (NFRI) funded by the government, the Slovenian Research Agency (ARRS) and NATO project SPS-984555. This work was also supported by Research and Business Development Program through the Korea Institute for Advancement of Technology (KIAT) funded by the Ministry of Trade, Industry and Energy (MOTIE) (Grant number: N0002038).

## Author contributions

S.P. and W.C. conceived the experiments. S.P. conducted all the experiments, and S.P., S.Y.M., and U.C. analyzed the results. W.C. supervised the work. S.P. and U.C. prepared the manuscript and figures, and all the authors contributed to the compilation and review of the manuscript.

## Additional information

**Competing interests:** The authors declare no competing financial interests.

