## [Peer Review File · Nature Communications]

Reviewers' comments:

Reviewer #1 (Remarks to the Author):

The paper presents the results of experimental study of generation of a neutral particle flow by the drag force imposed by charged particles. This phenomenon is important both for industrial devices using weakly ionised gases as well as for application to space physics.

This phenomenon was extensively studied theoretically. However, to my knowledge, the authors carried out its first careful experimental study. Both the device used for the experiment and the results obtained are described in detail. Although the paper is concise, all the necessary information is given.

My only concern is the language. Although, in general, it is not bad at all, there are some problems. Sometimes the authors use the single form of a noun instead of plural, or vice versa, or miss prepositions. For example, on line 333 it should not 'jet depends the density' but 'jet depends on the density'. I suggest that they check the language carefully.

Summarising, I am happy to recommend the paper for publication in Nature Communications.

Reviewer #2 (Remarks to the Author):

This paper calls attention to a phenomena by which an electric field can be used to influence the flow velocity of neutral atoms, an Electrohydrodynamic force: EHD. I struggled to understand what the new issue is being called out. The role of neutrals and forces between neutrals and charged particles have been studied extensively in many contexts. The ionospheric currents of the ionosphere, the role that neutrals play in coupling MHD waves to neutrals in astrophysics, where charge exchange and ionization couple the ion and neutrals and increase the overall mass density in the Alfvén speed, and in laboratory plasmas.

The novelty appears to be that the flow of the neutral gas is clearly measured to change due to the plasma being present and that this effect arises because the difference in drag between electrons and ions on the neutrals allows an electric field to persist--normally plasmas shield out these electric fields--due to lack of ambipolarity.

The advance appears to be that this system, and the time dependent way in which flows are measured, allows direct observation of the neutral flows.

I find the work interesting and convincing. The analysis technique is clearly laid out and the images of the plasma system taken by schlieren photography can be nicely related to the forces inferred.

The manuscript needs to be cleaned up grammatically. For example the sentences in lines 56-60 make no sense.

Response to Reviewers' Comments

NCOMMS-17-20517A

The authors appreciate the professional and valuable comments given by the reviewers. We are sure that the suggestions and comments will improve the quality of our manuscript.

Reviewer #1

- i. **The paper presents the results of experimental study of generation of a neutral particle flow by the drag force imposed by charged particles. This phenomenon is important both for industrial devices using weakly ionized gases as well as for application to space physics. This phenomenon was extensively studied theoretically. However, to my knowledge, the authors carried out its first careful experimental study. Both the device used for the experiment and the results obtained are described in detail. Although the paper is concise, all the necessary information is given.**

We have carefully addressed the reviewer's suggestions, and in doing so we feel the manuscript is substantially improved. The specific concerns have been addressed as follows.

- ii. **My only concern is the language. Although, in general, it is not bad at all, there are some problems. Sometimes the authors use the single form of a noun instead of plural, or vice versa, or miss prepositions. For example, on line 333 it should not ``\dots jet depends the density \dots' but ``\dots jet depends on the density \dots' I suggest that they check the language carefully.**

The reviewer was concerned with the language of our manuscript. First, we would like to make an apology that there was a language-related problem in the manuscript prepared for *Nature communications*, which is a high status journal. This is our critical mistake, due to miscommunications during the submission. As recommended by the editor, we used the editing service, *Nature Publishing Group Language Editing* (<http://languageediting.nature.com>). The main text and the supplementary information files were edited for English language usage, grammar, spelling and punctuation. Currently, the article is totally improved for clarity and readability. For example, we corrected all writing problems, including grammatical mistakes. As recommended by the reviewer, the expression “~ depends the density ~” is also corrected as “~ depends on the density ~”.

- iii. **Summarising, I am happy to recommend the paper for publication in Nature Communications.**

We thank the reviewer again for the comments and recommendations, and we hope that this revision clears the concern of the reviewer.

Reviewer #2

- i. **This paper calls attention to a phenomenon by which an electric field can be used to influence the flow velocity of neutral atoms, an Electrohydrodynamic force: EHD. I struggled to understand what the new issue is being called out. The role of neutrals and forces between neutrals and charged particles have been studied extensively in many contexts. The ion/pederson currents of the ionosphere, the role that neutrals play in coupling MHD waves to neutrals in astrophysics, where charge exchange and ionization couple the ion and neutrals and increase the overall mass density in the alfven speed, and in laboratory plasmas.**

The novelty appears to be that the flow of the neutral gas is clearly measured to change due to the plasma being present and that this effect arises because the difference in drag between electrons and ions on the neutrals allows an electric field to persist--normally plasmas shield out these electric fields--due to lack of ambipolarity.

The authors appreciate the professional and valuable comments. We have carefully addressed the reviewer's suggestions, and in doing so we feel the manuscript is substantially improved. The specific concerns have been addressed as follows:

- ii. **The advance appears to be that this system, and the time dependent way in which flows are measured, allows direct observation of the neutral flows.**

I find the work interesting and convincing. The analysis technique is clearly laid out and the images of the plasma system taken by schlieren photography can be nicely related to the forces inferred.

We would like to thank you for your thoughtful and detailed consideration of our work.

- iii. **The manuscript needs to be cleaned up grammatically. For example, the sentences in lines 56-60 make no sense.**

The reviewer pointed out the grammatical issues in the manuscript. First, we would like to make an apology that there was a language-related problem in the manuscript prepared for *Nature communications*, which is a high status journal. This is our critical mistake, due to miscommunications during the submission.

A problem in lines 56-60,

“In the jets, the horizontally ejected neutral helium flow has the flow trajectories or so-called free jet boundary bended upward as result of the buoyant force in ambient air. Faster is the flow, the lower in horizontal plane are bended the flow trajectories. However, when the plasma is turned on, the gas speed in jet increases and the flow trajectories become further horizontal along the flow too.”,

which was raised by the reviewer, is corrected as follows:

“In the jets, where a horizontally ejected neutral helium gas flow is used, the gas flow trajectories, or so-called free jet boundary, are typically bent upward due to the buoyant force. The flow

trajectories are lowered toward the horizontal plane when the neutral gas flow is faster. Nevertheless, when the electrical discharge, or so-called plasma, is generated inside the same flow, the gas speed of the flow increases. This is observed from the flow trajectories, which is additionally further lowered toward the horizontal plane of the flow.”

As recommended by the editor, we used the editing service, *Nature Publishing Group Language Editing* (<http://languageediting.nature.com>). The main text and the supplementary information files were edited for English language usage, grammar, spelling and punctuation. Currently, the article is totally improved for clarity and readability.

We thank the reviewer again for the comments and recommendations, and we hope that this revision will clear all the concern of the reviewer.

REVIEWERS' COMMENTS:

Reviewer #1 (Remarks to the Author):

I am completely satisfied with the author's response. Hence, Now I am happy to recommend the paper for publication.

Point-by-Point Response to Reviewers' Comments

In the following, we provide our replies to the reviewers' comments.

Reviewer #1:

- i. I am completely satisfied with the author's response. Hence, Now I am happy to recommend the paper for publication.**

The authors appreciate a reviewer's supporting comment.

Reviewer #2:

Reviewer #2 provided confidential comments, which support publication of our manuscript, to the editor.

We thank the reviewers again for the supporting comments and recommendations.